# VDAC1 negatively regulates melanogenesis through the Ca²⁺-calcineurin–CRTC1-MITF pathway

Jianli Wang[1],*, Juanjuan Gong[1],*, Qiaochu Wang[1], Tieshan Tang[2], Wei Li[1]

Melanocytes produce melanin for protecting DNA from ultraviolet exposure to maintain genomic stability. However, the precise regulation of melanogenesis is not fully understood. VDAC1, which is mainly localized in the outer mitochondrial membrane, functions as a gatekeeper for the entry or exit of Ca²⁺ between mitochondria and the cytosol and participates in multiple physiological processes. Here, we showed a novel role of VDAC1 in melanogenesis. Depletion of VDAC1 increased pigment content and up-regulated melanogenic genes, *TYR*, *TYRP1*, and *TYRP2*. Knockdown of VDAC1 increased free cytosolic Ca²⁺ in cultured melanocytes at the resting state, which activated calcineurin through the Ca²⁺-calmodulin-CaN pathway. The activated CaN dephosphorylated CRTC1 to facilitate its nuclear translocation and ultimately up-regulated the transcription of the master regulator of melanogenesis MITF. Consistently, depletion of *Vdac1* in mice led to up-regulation of the transcription of *MITF* and thereafter its targeted melanogenic genes. These findings suggest that VDAC1 is an important negative regulator of melanogenesis, which expands our knowledge about pigment production and implies its potential role in melanoma.

## Introduction

Melanin prevents UV-induced DNA damage by absorbing and scattering UVB radiation (Chiarelli-Neto et al, 2011). Melanogenesis is a complex process in which more than 150 genes are involved (Yamaguchi & Hearing, 2009). Abnormal melanin production may lead to hyper-/hypopigmented disorders or even life-threatening melanoma. However, the precise regulation of melanogenesis is very complicated. Thus, investigating the regulation of melanogenesis not only elucidates the mechanisms of pigmentary disorders but also provides new intervention clues to these diseases.

In melanocytes, there are multiple signaling pathways in regulating melanogenesis, among which the cAMP-PKA pathway plays a key role in this process. Induced by UVB or α-melanocyte–stimulating hormone (α-MSH), melanocortin 1 receptor (MC1R), the G protein-coupled receptors on the melanocyte membrane, is activated. In chain reaction, adenylate cyclase is activated, then cAMP and PKA (Kim et al, 2007; Yuan & Jin, 2018; D'Orazio & Fisher, 2011). Activated PKA enters into the nucleus to phosphorylate cAMP-response element–binding protein (CREB). It also phosphorylates salt-inducible kinase 2 (SIK2) in the cytoplasm, repressing the phosphorylation of cAMP-regulated transcriptional co-activator 1 (CRTC1) (Horike et al, 2010) and facilitating CRTC1 into the nucleus. However, p-CRTC1 in the cytoplasm binds to 14-3-3 protein which prevents its entrance into the nucleus (Altarejos & Montminy, 2011). Protein phosphatase 2B (PP2B), also known as calcineurin (Allouche et al, 2021), in the cytoplasm can dephosphorylate p-CRTC1, promoting the nuclear location of CRTC1 (Wadzinski et al, 1993; Bittinger et al, 2004). Consequently, CRTC1, p-CREB, and CREB-binding protein (CBP)/p300 in the nucleus cooperatively recruit the transcription machinery including RNA polymerase II to initiate the transcription of microphthalmia-associated transcription factor isoform M (MITF-M) and thus to promote melanogenesis (Hartman & Czyz, 2015; Yun et al, 2018). It has been shown that blocking the nuclear import of CRTC1 inhibits melanin production, suggesting that CRTC1 is a potential target in the treatment of pigmentary disorders (Yun et al, 2019).

Voltage-dependent anion channels (VDACs) are the most abundant proteins in the outer mitochondrial membrane, which are the barriers between the mitochondrial membrane and cytoplasm, controlling the transport of ATP and metabolite between mitochondria and the rest of the cell. VDAC1 is the most widely distributed protein of the VDAC family (Shoshan-Barmatz et al, 2018a). VDAC1 is a multifunctional channel mediating the entry of metabolites (e.g., NADH, pyruvate, malate, succinate, and nucleotides) into the mitochondria and the exit of newly formed molecules such as ROS into the cytosol (Shoshan-Barmatz et al, 2018b). VDAC1 also mediates Ca²⁺ to cross mitochondrial membrane (Shoshan-Barmatz et al, 2010; Shoshan-Barmatz & Ben-Hail, 2012) and regulates the permeability of Ca²⁺ in the inter-mitochondrial membrane (Gincel et al, 2001; Rapizzi et al, 2002; Tan & Colombini,

[1]Beijing Key Laboratory for Genetics of Birth Defects, Beijing Pediatric Research Institute; MOE Key Laboratory of Major Diseases in Children; Rare Disease Center, National Center for Children's Health; Beijing Children's Hospital, Capital Medical University, Beijing, China   [2]State Key Laboratory of Membrane Biology, Institute of Zoology, University of Chinese Academy of Sciences, Chinese Academy of Sciences, Beijing, China

Correspondence: liwei@bch.com.cn
*Jianli Wang and Juanjuan Gong contributed equally to this work.

2007). Thus, VDAC1 is generally accepted to be an important regulator in maintaining intracellular $Ca^{2+}$ homeostasis.

Several studies have shown that $Ca^{2+}$ in melanocytes is very important for melanogenesis (Carsberg et al, 1995; Zhang et al, 2019; Jia et al, 2020). $Ca^{2+}$ influx caused by SYT4 overexpression activates TRPM1 and MITF through CAMK4 (Jia et al, 2020). Melanin is synthesized within the melanosomes, a type of lysosome-related organelles (Li et al, 2022). Melanosome biogenesis/maturation is also regulated by $Ca^{2+}$ homeostasis in the melanosome. $Ca^{2+}$ enters the melanosome to regulate the maturation of melanosomes (Samuelson et al, 1993). Our previous study showed that NCKX5, located in the mitochondria, may play an important role in regulating melanosomal-$Ca^{2+}$ homeostasis that is required for pigment production (Zhang et al, 2019; Le et al, 2021). Thus, whether VDAC1 participates in the regulation of $Ca^{2+}$ homeostasis and melanogenesis in melanocytes is unknown.

In this study, we show that VDAC1 plays a novel role in regulating melanogenesis through the $Ca^{2+}$-calcineurin-CRTC1-MITF pathway, which provides insights into understanding the regulation of melanogenesis and potential intervention target for pigmentary disorders and melanoma.

# Results

## VDAC1 negatively regulates melanogenesis in melanocytes

To investigate whether VDAC1 has an effect on pigmentation, human melanoma MNT1 cells were transfected with siRNA targeting human *VDAC1*, and then melanin content was determined. The results showed that the melanin content was significantly increased after depletion of VDAC1 in MNT1 cells (Fig 1A–C). We further performed the same experiments in mouse skin melanocyte Melan-a cells. Likewise, knockdown of *Vdac1* led to increased melanin production compared with the negative controls (Fig 1D–F). These results suggest a novel role of VDAC1 in melanogenesis.

Melanosomes are organelles that synthesize, store, and transport melanin in melanocytes (D'Alba & Shawkey, 2019). We explored whether melanosomes are compromised with VDAC1 deficiency. We used transmission electron microscope (TEM) to observe the melanosomes in MNT1 cells. The number of melanosomes per melanocyte was apparently increased after knockdown of VDAC1 (Fig 1G and H), suggesting that depletion of VDAC1 affects the homeostasis of melanosomes, which may lead to the increase of melanin production.

## VDAC1 regulates the expression of melanogenic proteins

In mammals, pigments are produced in melanosomes, where key melanogenic proteins were recruited to synthesize melanin, such as tyrosinase (TYR), tyrosinase-related protein 1 (TYRP1), and dopachrome tautomerase (TYRP2/DCT) (D'Alba & Shawkey, 2019). Western blotting showed that the expression levels of TYR, TYRP1, and TYRP2 were up-regulated in siRNA-*VDAC1* MNT1 cells compared with the siRNA-NC cells (Fig 2A–D). Consistently, in Melan-a cells, knockdown of *Vdac1* increased the protein level of TYR, TYRP1, and TYRP2 (Fig 2E–H). TYR is the rate-limiting step in melanin biosynthesis

(Hearing, 2011). Quantitative RT–PCR (qRT-PCR) was conducted to show that the mRNA level of *TYR* increased in the VDAC1 knockdown group (Fig 2I). These results suggest that VDAC1 regulates the key proteins of melanogenesis at the transcriptional level.

## Knockdown of VDAC1 in melanocytes promotes the CREB/CRTC1-MITF pathway

In melanocytes, MITF-M is the transcription factor of *TYR*, *TYRP1*, and *TYRP2* (Hartman & Czyz, 2015), which is a master regulator of melanin synthesis and melanosome biogenesis. As *TYR* is transcriptionally up-regulated by depletion of VDAC1, we tested whether VDAC1 knockdown affects the level of MITF. Our results showed that VDAC1 knockdown increased the protein level of MITF (Fig 3A and B). qRT-PCR data further confirmed that the mRNA level of *MITF* was increased after VDAC1 depletion (Fig 3C).

To explore how MITF is up-regulated upon VDAC1 knockdown, the phosphorylated-CREB (p-CREB) and CRTC1, which are important for the transduction of the cAMP signal to induce the transcription of MITF, were examined (Kim et al, 2019). p-CREB and CRTC1 were detected by Western blotting after nuclear and cytoplasmic proteins were isolated in MNT1 cells. Our results showed that the protein level of CRTC1 in the nucleus increased, whereas there was no significant difference in the p-CREB level between the VDAC1 knockdown group and control group (Fig 3D–F). These results suggest that depletion of VDAC1 up-regulates the transcription of MITF through enhancing the nuclear translocation of CRTC1, which in turn increases the expression of *TYR*, *TYRP1* and *TYRP2*.

## Knockdown of VDAC1 activates the $Ca^{2+}$-calmodulin-calcineurin pathway

It has been shown that dephosphorylation of CRTC1 can lead to the translocation of CRTC1 into the nucleus and enhance the binding of p-CREB to its CRE domain in the promoter region (Altarejos & Montminy, 2011). In melanocytes, SIK2 phosphorylates CRTC1 and CaN dephosphorylates CRTC1. CaN is a highly conserved serine/threonine protein phosphatase that can be specifically activated by $Ca^{2+}$-calmodulin (CaM). It couples intracellular $Ca^{2+}$ signaling and cell response, which is important for multiple cellular functions (Klee et al, 1979). After increasing intracellular $Ca^{2+}$, CaM binds $Ca^{2+}$ to recruit the catalytic subunit of CaN, and then CaN dephosphorylates its substrates (Fox & Heitman, 2002). As SIK2 and p-CREB are both activated by cAMP (Horike et al, 2010), the unchanged p-CREB level (Fig 3F) suggests that the cAMP-PKA pathway might not participate in the regulation of melanogenesis under this resting condition without UVB or $α$-MSH. In addition, VDAC1 mediates mitochondrial $Ca^{2+}$ influx or efflux. We hypothesized that the increased nuclear translocation of CRTC1 caused by VDAC1 depletion may be because of its dephosphorylation through the activation of CaN triggered by the $Ca^{2+}$-CaM-CaN pathway.

It has been reported that VDAC1 knockdown in cells significantly reduces mitochondrial $Ca^{2+}$ influx induced by histamine or $H_2O_2$, and overexpression of VDAC1 increases mitochondrial $Ca^{2+}$ uptake (Madesh & Hajnoczky, 2001; De Stefani et al, 2012). We therefore reasoned that knockdown of VDAC1 might increase resting $Ca^{2+}$ in the cytoplasm by preventing its entry into the mitochondria. To verify this point, the mKate-linker-GCaMP6m plasmid (miG6m) was

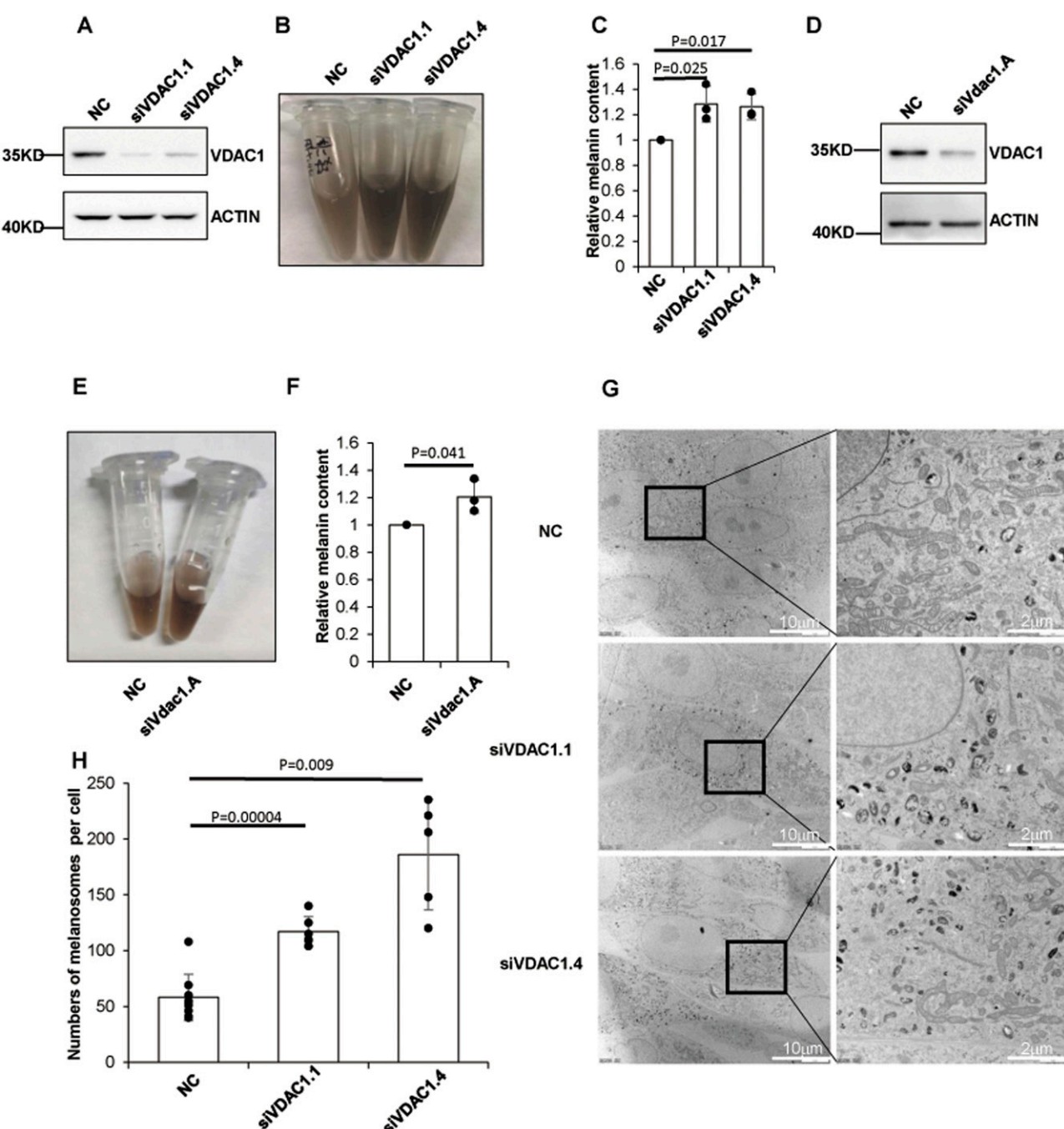

**Figure 1.  VDAC1 regulates melanogenesis.**
**(A, B, C)** Depletion of VDAC1 increases the production of melanin in MNT1 cells. MNT1 cells were harvested after transfected with two different siRNA targeting human *VDAC1* for 96 h. **(B, C)** Melanin was extracted and determined as optical density (OD) at 450 nm (B, C). Data are mean ± SD from three independent experiments. **(A)** Efficiency of siRNA targeting VDAC1 was confirmed by Western blotting (A). **(D, E, F)** Depletion of VDAC1 increases the production of melanin in Melan-a cells. Melan-a cells transfected with siRNA targeting mouse *Vdac1* for 96 h were lysed. **(E, F)** Melanin was extracted and determined (E, F). Data are mean ± SD from three independent experiments. **(D)** Efficiency of siRNA targeting *Vdac1* was confirmed by Western blotting (D). **(G, H)** VDAC1 knockdown increases the number of melanosomes. MNT1 cells with siRNA targeting *VDAC1* for 96 h were collected for detecting the number of melanosomes per cell under the electron microscope (the cell number of each group >5).

used to monitor the concentration of $Ca^{2+}$ in the cytosol. The results showed that the concentration of resting $Ca^{2+}$ increased significantly in MNT1 cells after VDAC1 knockdown (Fig 4A). Because cytoplasmic free $Ca^{2+}$ is involved in a variety of biological activities, this change of the

basal cytoplasmic $Ca^{2+}$ level in si*VDAC1* cells may influence several processes through regulating the $Ca^{2+}$ signaling pathway.

In melanocytes, increased $Ca^{2+}$ in the cytosol may activate CaN, which leads to the dephosphorylation of CRTC1 and eventually

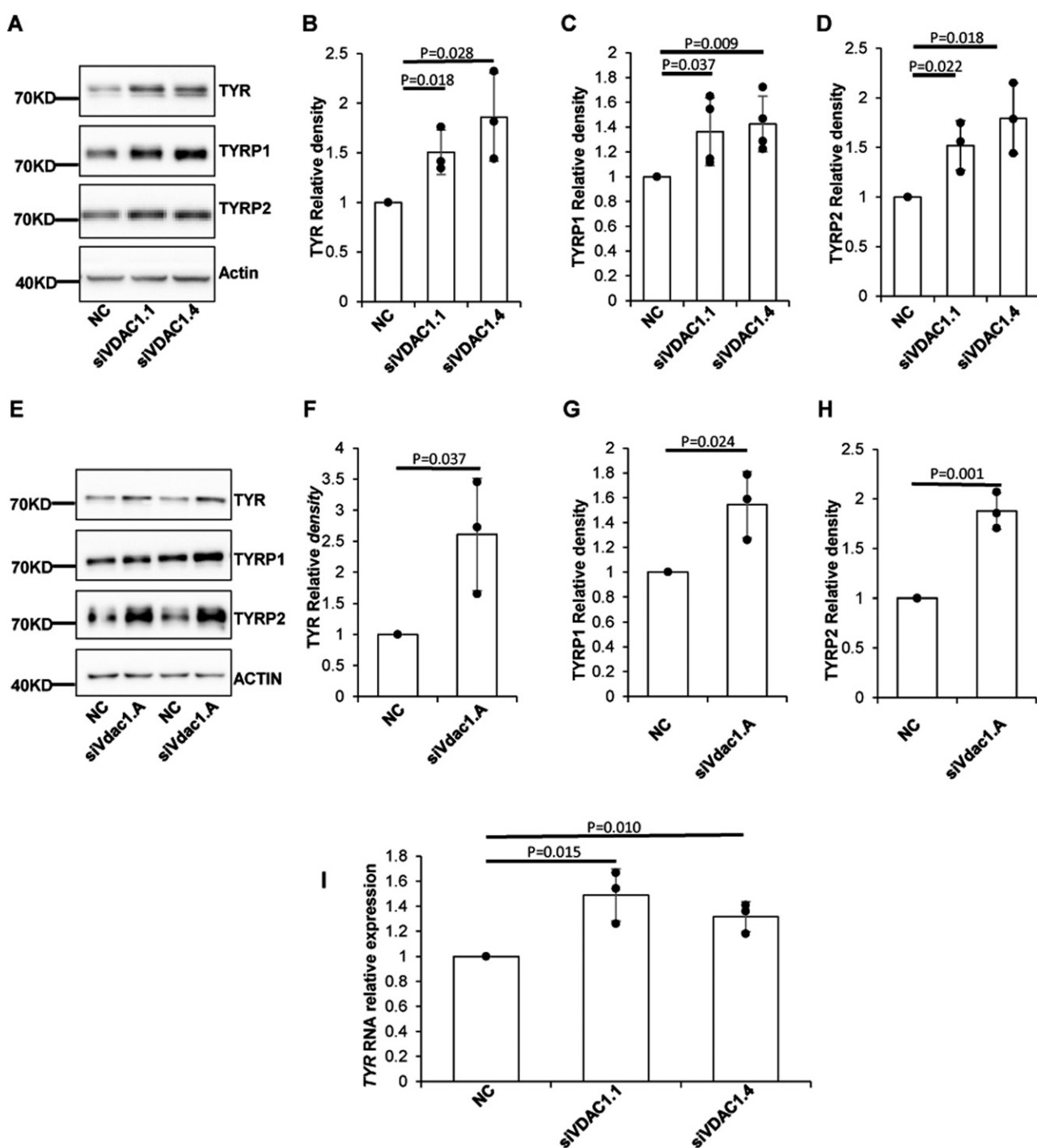

**Figure 2. The effect of VDAC1 on melanogenic genes.**
**(A, E)** Depletion of VDAC1 up-regulates the expression of the melanogenic proteins (TYR, TYRP1, TYRP2) in MNT1 cells (A) or Melan-a cells (E). Cells transfected with siRNA targeting human *VDAC1* or mouse *Vdac1* for 72 h were lysed and subjected to Western blotting analysis with anti-tyrosinase, anti-TYRP1, anti-TYRP2, or anti-GAPDH antibody. **(B, C, D)** Depletion of VDAC1 up-regulates the expression of the melanogenic proteins in MNT1 cells. Western blotting analyses of the relative expression of TYR, TYRP1, TYRP2 in NC and *VDAC1* knockdown MNT-1 cells. Data are mean ± SD from three independent experiments. **(F, G, H)** Statistical analysis of the relative expression of TYR, TYRP1, TYRP2 in NC and *Vdac1* knockdown Melan-a cells. Data are mean ± SD from three independent experiments. **(I)** Knockdown of *VDAC1* increases the expression of *TYR* transcriptionally. Total RNA was extracted after MNT1 cells were transfected with siRNA for 72 h, and the mRNA level of *TYR* was determined by quantitative RT-PCR. Data are mean ± SD from three independent experiments.

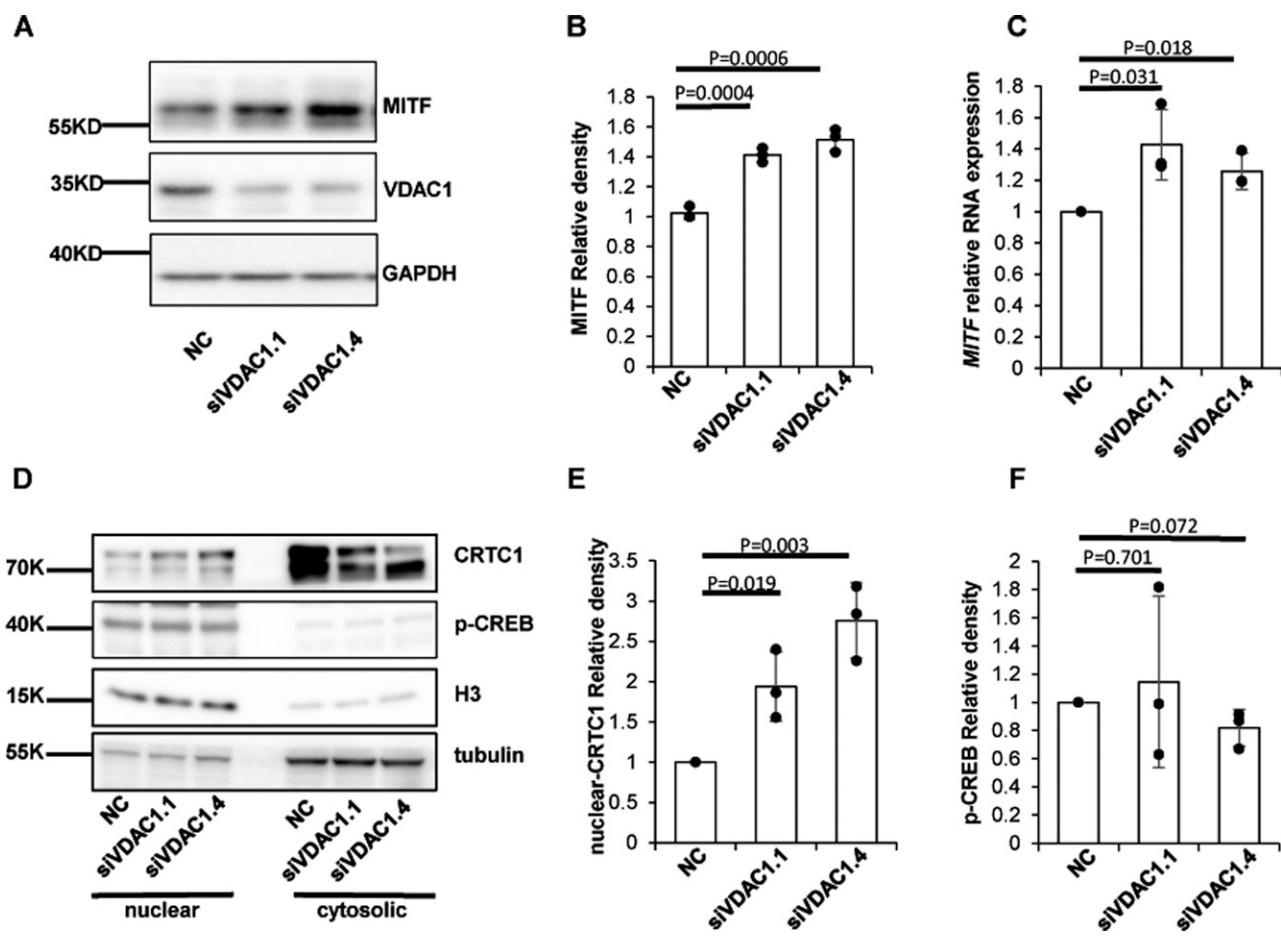

**Figure 3. VDAC1 is involved in the CRTC1-MITF pathway.**
**(A, B)** Knockdown of *VDAC1* increases the expression of MITF. MNT1 cells with siRNA targeting *VDAC1* for 48 h were lysed and subjected to Western blotting with antibody to MITF. Data are mean ± SD from three independent experiments. **(C)** The expression level of *MITF* mRNA in NC and *VDAC1* knockdown cells. MNT1 cells were transfected with siRNA-*VDAC1* for 48 h. The level of *MITF* mRNA was detected by quantitative RT-PCR. The relative expression of *MITF* RNA (relative to GAPDH) was calculated. Data are mean ± SD from three independent experiments. **(D, E, F)** Depletion of VDAC1 promotes the nuclear translocation of CRTC1. Nuclear proteins and cytosolic proteins were separated after MNT1 cells were transfected with siRNA-*VDAC1* for 48 h. The expression of CRTC1 and p-CREB were detected by Western blotting. H3 protein serves as a reference for nuclear protein and tubulin as a reference for cytosolic protein. Data are mean ± SD from three independent experiments.

increases the nuclear translocation of CRTC1. To verify whether inhibition of CaN may interfere the nuclear localization of CRTC1 and subsequently suppress the melanin production, CsA, a CaN inhibitor, was used to incubate with the MNT1 cells. Our results showed that in the CsA treatment group, the content of melanin and the expression of TYR were decreased compared with the control cells (Fig 4B–E). In addition, the nuclear translocation of CRTC1 was reduced by CsA (Fig 4F and G). Furthermore, CsA markedly inhibited melanogenesis induced by VDAC1 knockdown (Fig 4H). These results suggest that CaN regulates the CRTC1-MITF pathway in response to resting cytoplasmic Ca²⁺ and that the regulation of melanogenesis by VDAC1 is dependent on Ca²⁺-CaM-CaN activation.

### VDAC1 regulates melanogenesis in vivo

To explore the effect of VDAC1 on pigmentation in vivo, *Vdac1*-knockout mouse line (*Vdac1⁻/⁻*) was successfully generated using the CRISPR-Cas9 technology in the C57BL/6J background (Fig 5A–D). The

number of melanosomes in the eyes of *Vdac1⁻/⁻* mice was increased, but the size of melanosomes was decreased compared with that in *Vdac1⁺/⁺* (wild-type, WT) mice by TEM (Fig 6 A–C). Consistent with the qPCR data in vitro, the mRNA levels of *Mitf*, *Tyr*, *Tyrp1*, and *Tyrp2* were increased in *Vdac1⁻/⁻* mice (Fig 6D–G). These confirmed that VDAC1 regulates the expression of melanogenic genes in vivo. However, there was no obvious change in melanin content (Fig 5E and F) and the melanocyte number between *Vdac1⁻/⁻* mice and controls (Fig 7A and B). The complex signals that affect melanogenesis in vivo, such as the endogenous hormone α-MSH, may cover up the effect caused by *Vdac1* depletion. Thus, our work provides a new melanogenic mechanism in a UVB- or α-MSH–independent manner.

## Discussion

VDAC1 transfers Ca²⁺ to the mitochondria. When VDAC1 is deficient, Ca²⁺ may be accumulated in the cytoplasm because of the

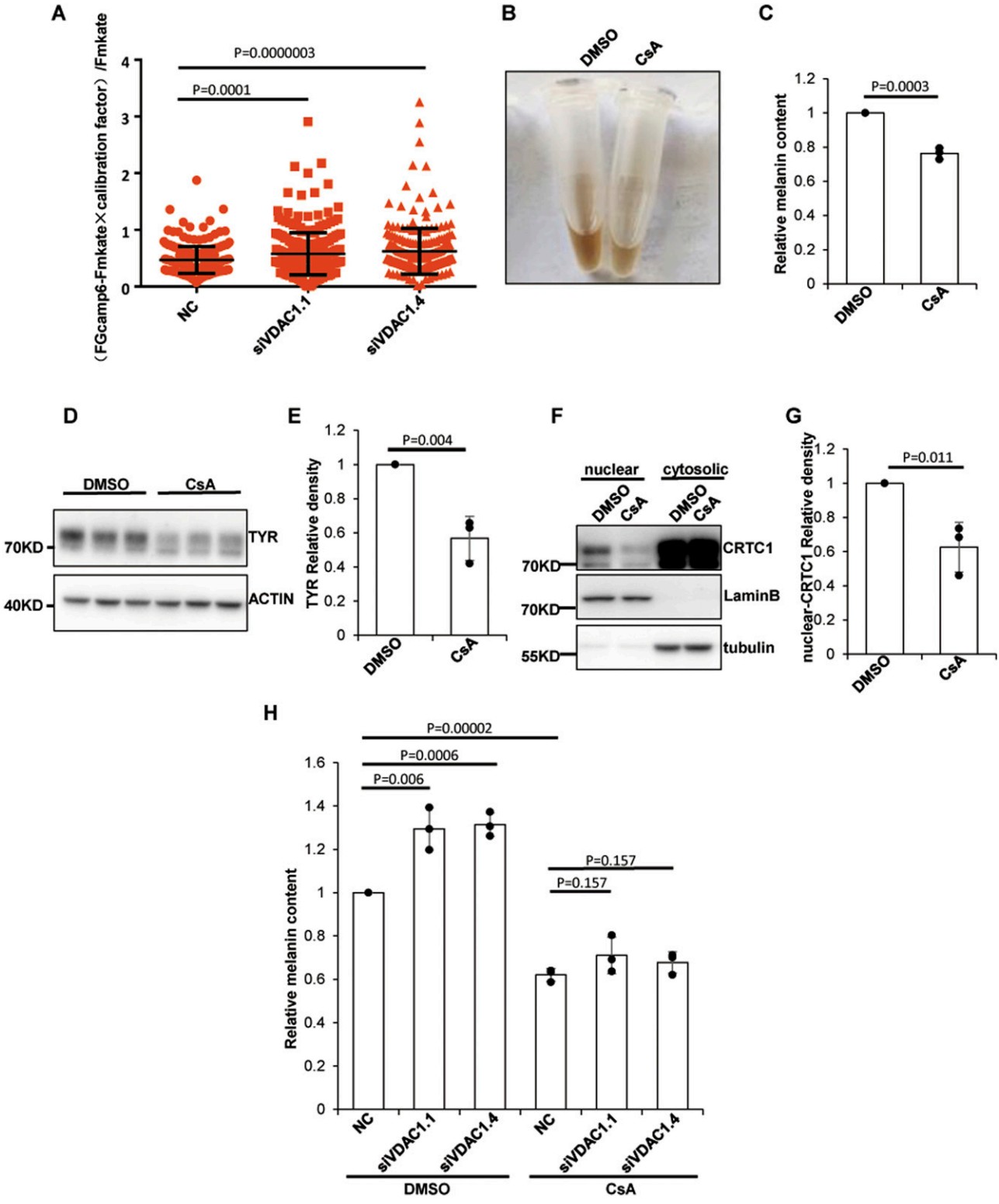

**Figure 4. VDAC1 regulates melanogenesis through the Ca²⁺-calcineurin-CRTC1 pathway.**
**(A)** Depletion of VDAC1 increases the basal $Ca^{2+}$ level. The mKate-linker-GCaMP6m plasmid was transfected to MNT1 cells 48 h after pre-transfected with siRNA targeting *VDAC1*. The fluorescence intensity of GCaMP6m and mKate in the cytoplasm of each cell was detected by fluorescence microscopy. Data plotted are from two independent experiments. The number of cells counted in the NC group were 252; in the si*VDAC1*.1 group, 280; and in the si*VDAC1*.4 group, 253. Data are shown are mean ± SEM. **(B, C)** A calcineurin inhibitor, CsA, suppresses the production of melanin. MNT1 cells were incubated with 10 μM CsA for 72 h, and then melanin was extracted and measured at OD450. Data are mean ± SD from three independent experiments. **(D, E)** Suppression of calcineurin inhibited the expression of TYR. MNT1 cells preincubated with 5 μM CsA

inefficient uptake of $Ca^{2+}$ by the mitochondria; therefore, free $Ca^{2+}$ in the cytoplasm is increased under the resting condition. $Ca^{2+}$ in the cytoplasm binds to CaM to increase its activity through the $Ca^{2+}$-CaM-CaN pathway. CaN dephosphorylates p-CRTC1 to disassociate CRTC1 from 14-3-3 protein and promotes the nuclear translocation of CRTC1. After CRTC1 enters the nucleus, it initiates the transcription of MITF together with p-CREB. Ultimately, the expression of key proteins for melanogenesis in melanocytes such as TYR, TYRP1, and TYRP2 are increased, and melanosome biogenesis could be up-regulated upon VDAC1 knockdown (Fig 7C). Except for the well-known UVB/α-MSH-cAMP-PKA pathway in regulating melanogenesis, our results reveal that the $Ca^{2+}$-CaM-CaN pathway is important for the production of a basal level melanin under resting condition, in which VDAC1 is an important negative regulator of melanogenesis. In addition, the aberrant melanin production because of abnormal regulation of VDAC1 may be implicated in pathological hyper- or hypo-pigmentation, such as albinism, freckles, or even life-threatening melanoma. Our work provides a potential target for these disorders.

In cultured melanocytes, we have shown that VDAC1 negatively regulates melanogenesis through the $Ca^{2+}$-calcineurin-CRTC1-MITF pathway. Although there was no significant difference in the pigment content between $Vdac1^{-/-}$ mice and controls, the increased number of melanosomes and increased transcription of key proteins in melanogenesis were indications of the up-regulation of pigmentation. One possible explanation is that the $Ca^{2+}$-CaN-CRTC1-MITF pathway plays a pivotal role in melanogenesis in cultured melanocytes without UVB or α-MSH stimulation. In the mice, alternative pathways such as the α-MSH-cAMP-PKA pathway mainly substitute for the production of melanin, which may cover up the effects of the disrupted $Ca^{2+}$-CaN-CRTC1-MITF pathway on melanin production in vivo. Likewise, the increased number of melanosomes in RPE may also be compensated by the decreased size of melanosomes, without much increase in melanin content, although the melanogenic genes are up-regulated. In addition, the $Vdac1^{-/-}$ mice were generated in the C57BL/6J background. The C57BL/6J mice are black, which may make it difficult to distinguish whether the fur color of mice becomes much darker when $Vdac1$ is deficient. The inbred strain mice with agouti appearance may be another choice. The $Vdac1^{-/-}$ mice in the C57BL/6J background will be transferred to the agouti background to observe possible color changes in vivo in our future work.

Intracellular $Ca^{2+}$ homeostasis participates in many cell activities, such as oxidative phosphorylation, $Ca^{2+}$ signaling, cell death, and secretion and production of ROS (Berridge et al, 2003). It has been shown that $Ca^{2+}$ can influence melanogenesis in melanocytes. Beyond the $Ca^{2+}$-CaN-CRTC1-MITF pathway, other effects of cytoplasmic $Ca^{2+}$ increase may exist. (1) PMEL, a key structural protein in the development of melanosomes, is cleaved by proprotein convertase (PC) (Hurbain et al, 2008; Hoashi et al, 2010). The activity of PC depends on $Ca^{2+}$ concentration (Thomas, 2002). Therefore, intracellular $Ca^{2+}$ may affect the melanosomal $Ca^{2+}$ for the maturation of melanosomes through regulating PC activity. (2) It has been demonstrated that protein kinase C (PKC)-β activates TYR activity through phosphorylation of TYR and participates in melanogenesis, whereas cytoplasmic $Ca^{2+}$ can affect PKC-β activity (Park et al, 1993, 2006; Newton, 2018). Whether these pathways play a role in VDAC1 deficiency requires further investigation.

Mitochondria play a pivotal role in melanogenesis in different manner. (1) Melanosomes are physically tethered to mitochondria, and this connection is important for melanosome biogenesis (Daniele et al, 2014). The melanosome–mitochondrion contact frequency appears to correlate positively with the process of melanosome biogenesis, suggesting that these contacts possibly facilitate early stages of melanosome maturation by ATP and/or $Ca^{2+}$ supply (Wu & Hammer, 2014; Zhang et al, 2019). (2) Mitochondrial fusion enhances melanin synthesis, whereas mitochondrial fission inhibits hyper-pigmentation because of down-regulation of melanogenic enzymes through the ROS-ERK signaling pathway (Kim et al, 2014). (3) Redox regulates melanogenesis by regulating tyrosinase protein stability and melanosome maturation via the enzyme nicotinamide nucleotide transhydrogenase, located to the inner mitochondrial membrane, which can regulate mitochondrial redox levels (Allouche et al, 2021). Our research provides a new angle that the mitochondrial protein VDAC1 can regulate melanogenesis by regulating cytosolic $Ca^{2+}$ to activate the $Ca^{2+}$-CaN-CRTC1-MITF pathway. All these lines of evidence support the importance and complexity of mitochondria in melanogenesis.

In addition, VDAC1 has multiple functions in regulating metabolism, cell growth, proliferation, and differentiation (Arif et al, 2017). They pointed out that 4,493 genes were significantly changed after the knockdown of VDAC1. The down-regulated genes were mainly those genes related to cell cycle, DNA repair, and mitochondrial function, whereas the up-regulated genes were mainly the genes related to transcription factors and the Wnt signaling pathway. Consistently, our study showed that depletion of VDAC1 up-regulated the expression of MITF transcriptionally. VDAC1 has been shown a proapoptotic or autophagic factor in melanoma or other cancers. VDAC1 knockdown inhibits apoptosis or autophagy of melanoma cells (Wang et al, 2014; Seo et al, 2019). We here showed that VDAC1 knockdown promotes melanogenesis which may also contribute to the development of melanoma.

In conclusion, our work showed that VDAC1 regulates melanin production in an α-MSH/UVB–independent manner, which may provide a potential target for pigmentary disorders and melanoma.

for 72 h were lysed and subjected to Western blotting with antibody to TYR. Data are mean ± SD from three independent experiments. **(F, G)** Suppression of calcineurin attenuates the nuclear localization of CRTC1. Nuclear proteins and cytosolic proteins were separated from MNT1 cells preincubated with 5 μM CsA for 72 h. The protein level of CRTC1 was detected by Western blotting. Lamin B protein served as a reference for nuclear protein and tubulin as a reference for cytosolic protein. Data are mean ± SD from three independent experiments. **(H)** Calcineurin inhibitors diminish the increase in melanin content caused by VDAC1 depletion. MNT1 cells were seeded in medium with 5 μM CsA and harvested 48 h after transfected with siRNA-NC/*VDAC1*, and the melanin was extracted and measured at OD450. Data are mean ± SD from three independent experiments.

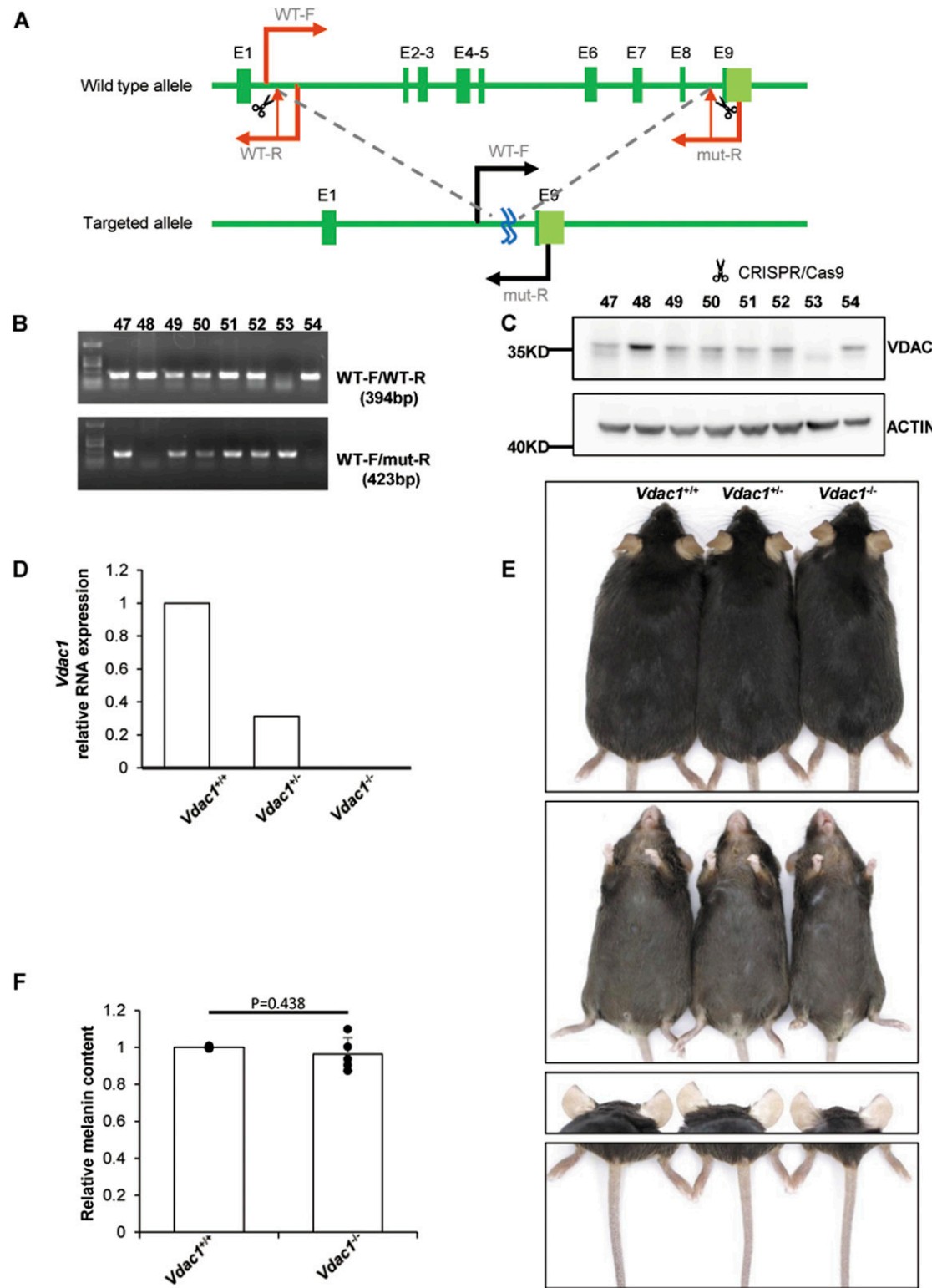

**Figure 5. *Vdac1*-knockout mouse line is generated using CRISPR-Cas9 in the C57BL/6J.**
**(A)** The scheme of *Vdac1*-knockout strategy. Exons 2~8 are targeted by CRISPR/Cas9 technology. *Vdac1*-WT-F, *Vdac1*-WT-R, and *Vdac1*-mut-R indicate the locations of the primers used for genotyping. **(B)** Verification of *Vdac1* deletion at genomic level. Mice are numbered as #47~#54. The upper bands are amplicons with primers *Vdac1*-WT-F and *Vdac1*-WT-R (394 bp). These bands can only be amplified in wild-type or heterozygous mice and cannot be amplified in homozygous mice because of the absence of the sequence of primer *Vdac1*-WT-R. The lower bands show the amplicons of primers *Vdac1*-WT-F and *Vdac1*-mut-R (423 bp). These bands can only be amplified in heterozygous and homozygous mice, and the sequence between *Vdac1*-WT-F and *Vdac1*-mut-R is too long to be amplified in wild-type mice. According to the results, #53

# Materials and Methods

### Cell culture and transfection

MNT1 cells were from Prof. Tiechi Lei (Renmin Hospital of Wuhan University) and grown in MEM medium (SH30024.01; HyClone) supplemented with 20% fetal bovine serum (10099-141; GIBCO), 10% AIM-V+AlbuMAX (31035025; GIBCO), 1% sodium pyruvate (11360070; GIBCO), and 1% Non-Essential Amino Acids Solution (100×) at 37°C and 5% $CO_2$. Melan-a cells were from the laboratory of Dorothy C. Bennett (St. George's University) (Bennett et al, 1987) and grown in DMEM medium (11965092; GIBCO) supplemented with 10% fetal bovine serum (10099-141; GIBCO) at 37°C and 10% $CO_2$. siRNAs (Table S1) were transfected into cells to knock down *VDAC1* using Lipofectamin RNAiMAX Transfection Reagent (13778100; Invitrogen) according to the manufacturer's instructions.

### Melanin measurement

Cultured cells and tissue samples were lysed in a lysis buffer (1% NP40, 0.9% NaCl, 5% 1 M Tris–HCl, pH 7.4) for 30 min, then centrifuged at 12,000*g* for 15 min at 4°C. The sediment was completely dissolved in the melanin eluent (8% NaOH, 20% DMSO) at 60°C for 1 h. Then OD450 of each sample was detected using the microplate reader (Multiskan Go; Thermo Fisher Scientific), and the values of OD450 were normalized by protein concentration.

### Immunoblotting

Cells and tissue samples were lysed in a lysis buffer as described above, then centrifuged at 12,000*g* for 15 min at 4°C, and the supernatants were used for Western blotting. Nuclear proteins were isolated from cells using a nuclear and cytoplasmic protein extraction kit (P0027; Beyotime). Proteins were separated by SDS–PAGE gels and transferred to PVDF membranes (Millipore). The membranes were blocked with 5% nonfat dried milk in Tris-buffered saline with 0.1% Tween 20 for 1 h. Then the membranes were incubated with specific primary antibodies (details in Table S2) at 4°C overnight and followed by incubation with secondary antibodies. The blots were detected using the ECL chemiluminescence reagent (A38555; Thermo Fisher Scientific).

### Quantitative real-time PCR

RNAs were isolated from cells and tissue samples using the RNeasy Mini Kit (74104; QIAGEN). The concentrate of RNA in each sample was measured using a spectrophotometer (DS-11; DeNovix). cDNA was reverse-transcribed by the iScript cDNA Synthesis Kit (1708890; Bio-Rad). Quantitative real-time PCR was performed in a Rotor-Gene Q (QIAGEN), using SuperReal PreMix Color SYBR Green (FP215; TIANGEN) and corresponding primers (Table S3). Two duplications were set up in each sample. The cycles to threshold (CT) were measured for each well, normalized to GAPDH, and calculated by $2^{-\Delta\Delta CT}$.

### Resting Ca²⁺ measurement

After knockdown of *VDAC1* in MNT1 cell, miG6m (mKate-linker-GCaMP6m) provided by Prof. Youjun Wang from Beijing Normal University was transfected into MNT1 cells. GCaMP6m (green) fluorescence intensity reflects the $Ca^{2+}$ level, and mKate (red) fluorescence intensity reflects the protein level of miG6m. Images were taken by a total internal reflection fluorescence microscope (DMI6000B; Leica). The mKate plasmid was expressed alone in cells, stimulated by green and red lights, respectively, and $F_{GFP}/F_{mKate}$ was used for calibration factor. Finally, the levels of resting $Ca^{2+}$ = $(F_{Gcamp6} - F_{mkate} \times calibration factor)/F_{mkate}$.

### Transmission electron microscopy

For TEM analysis, MNT1 cells transfected with siRNAs for 48 h were fixed with 2.5% (vol/vol) glutaraldehyde with phosphate buffer (0.1 m, pH 7.4) and washed four times with phosphate buffer at 4°C. For the eyes of mice, samples were rinsed with 0.1 M phosphate buffer saline (pH 7.2) and placed in 2.5% glutaraldehyde for 8 min. The cornea was cut, the lens was extruded, and then the samples were kept in 2.5% glutaraldehyde overnight. Samples were postfixed with 1% (wt/vol) osmium tetroxide ($OsO_4$) and 1.5% (wt/vol) potassium ferricyanide aqueous solution at 4°C for 2 h. Samples were dehydrated through a graded ethanol series (30, 50, 70, 80, 90, 100, and 100%, 5 min each at 4°C) into pure acetone (2 × 5 min), infiltrated in a graded mixture (3:1, 1:1, 1:3) of acetone and SPI-PON812 resin (16.2 ml of SPI-PON812, 10 ml of dodecanoyl succinic anhydride and 8.9 ml of methyl nadic anhydride), and then changed to pure resin. Finally, samples were embedded in pure resin with 1.5% N, N-dimethylbenzylamine and polymerized for 12 h at 45°C and 48 h at 60°C. The ultrathin sections (70-nm thick) were prepared with a microtome (EM UC7; Leica), double-stained by uranyl acetate and lead citrate, and imaged by a TEM (FEI Tecnai Spirit 120 kV).

### Immunofluorescence confocal imaging

The mouse dorsal skin was formalin-fixed and paraffin-embedded, and 5-$\mu$m slices of the skin were sectioned with a rotary paraffin microtome (RM2255; Leica) and mounted on positively charged slides. Then slices were steamed for 20 min in 0.01 M sodium citrate buffer (pH 6.0) for antigen retrieval, blocked with 1% normal goat serum (AR0009, Wuhan booster) in PBS buffer for 30 min at room temperature, incubated in primary antibody solution (anti-DCT 1: 200, DCT antibodies were kindly gifted from Dr. Ting Chen's Lab,

is *Vdac1⁻/⁻*, #48 and #54 are *Vdac1⁺/⁺*, and all the others are *Vdac1⁺/⁻*. **(C)** Verification of *Vdac1* deletion at the protein level. **(D)** Verification of *Vdac1* deletion at the mRNA level. The level of *Vdac1* mRNA extracted from the whole eye was detected by quantitative PCR. The relative mRNA expression of *Vdac1* (relative to GAPDH) was calculated. **(E)** No apparent color change in the dorsal fur, ventral fur and toes, ears, and tail of *Vdac1⁻/⁻* or *Vdac⁺/⁻* mouse was observed compared with the wild-type mouse. **(F)** The whole eye of mouse was extracted by lysis, and the melanin content was determined at OD450. The melanin content of wild-type mice in the same group was standardized. Data shown are mean ± SD from five independent experiments.

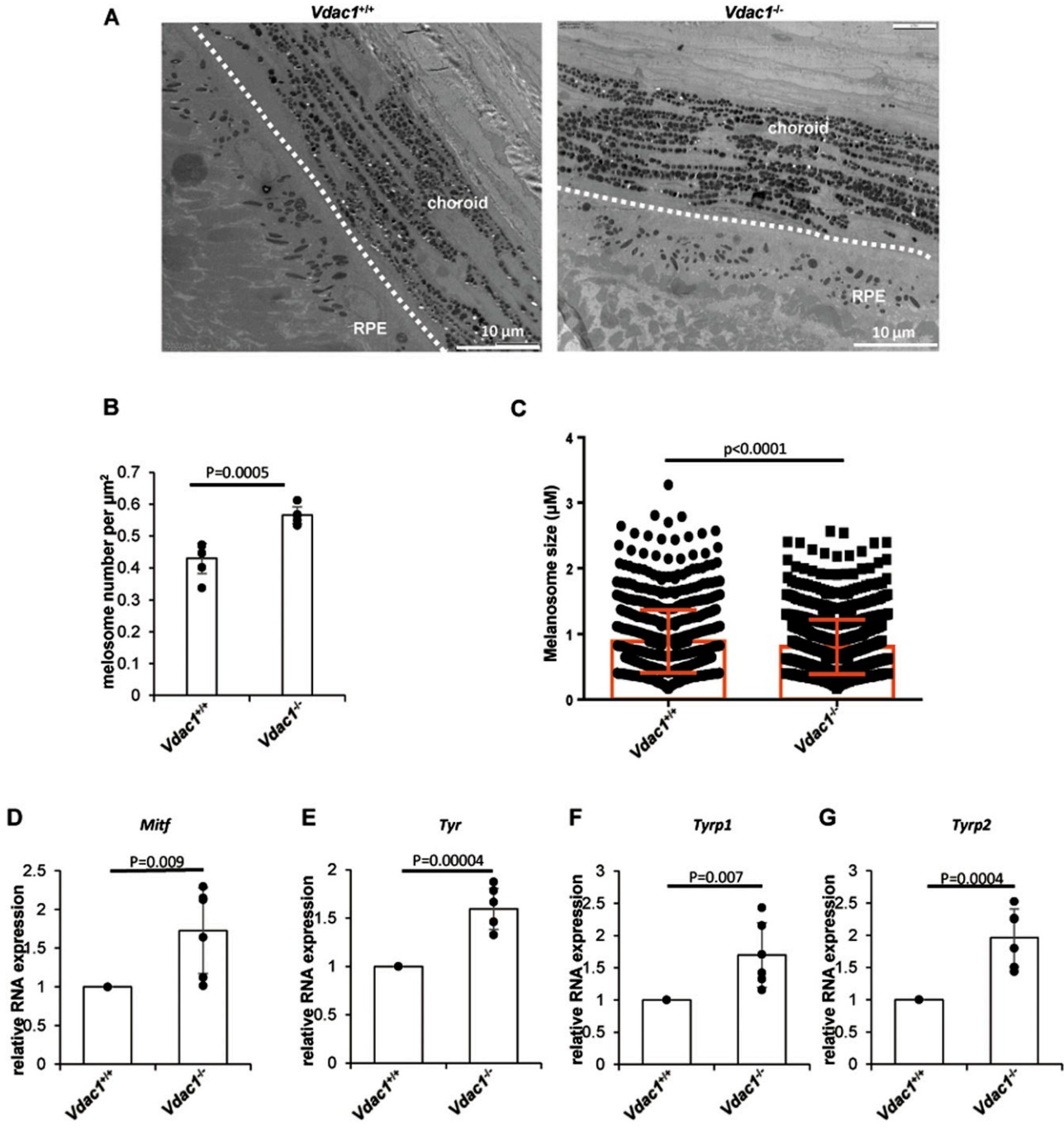

**Figure 6. Melanogenesis is up-regulated in *Vdac1*-knockout mice.**
**(A, B, C)** The number and size of melanosomes of the gross area of retinal pigment epithelium by the transmission electron microscope are shown (A). **(B)** The melanosome number per $\mu m^2$ was calculated, which was increased in *Vdac1*$^{-/-}$ mice compared with WT mice; data shown are mean ± SD (B). The size of melanosome in *Vdac1*$^{-/-}$ mice was decreased compared with WT mice. The number of melanosomes: WT, 850; *Vdac1*$^{-/-}$, 1,025. **(C)** Data shown are mean ± SEM (C). Scale bar: 10 $\mu m$. **(D, E, F, G)** Transcription of key proteins in melanogenesis is increased in *Vdac1*-KO mice. Total RNAs from the eyes of male mice were extracted, and mRNA levels of *Mitf* and three key genes of melanogenesis (*Tyr*, *Tyrp1*, *Tyrp2*) were determined by quantitative PCR. Data are mean ± SD from four independent experiments.

National Institute of Biological Sciences) at 4°C overnight, washed with PBS buffer for three times, and incubated in secondary antibody solution (1:500). They were mounted with mounting medium with DAPI. Confocal images were acquired with a 10× objective with NA 0.45 on a Zeiss LSM 880 confocal microscope (Nikon).

**Fontana-Masson staining**

Fontana-Masson staining was performed with the Fontana-Masson Stain Kit (Solarbio Life Sciences). Briefly, skin slices were deparaffinized and hydrated in distilled water. Slices were incubated in

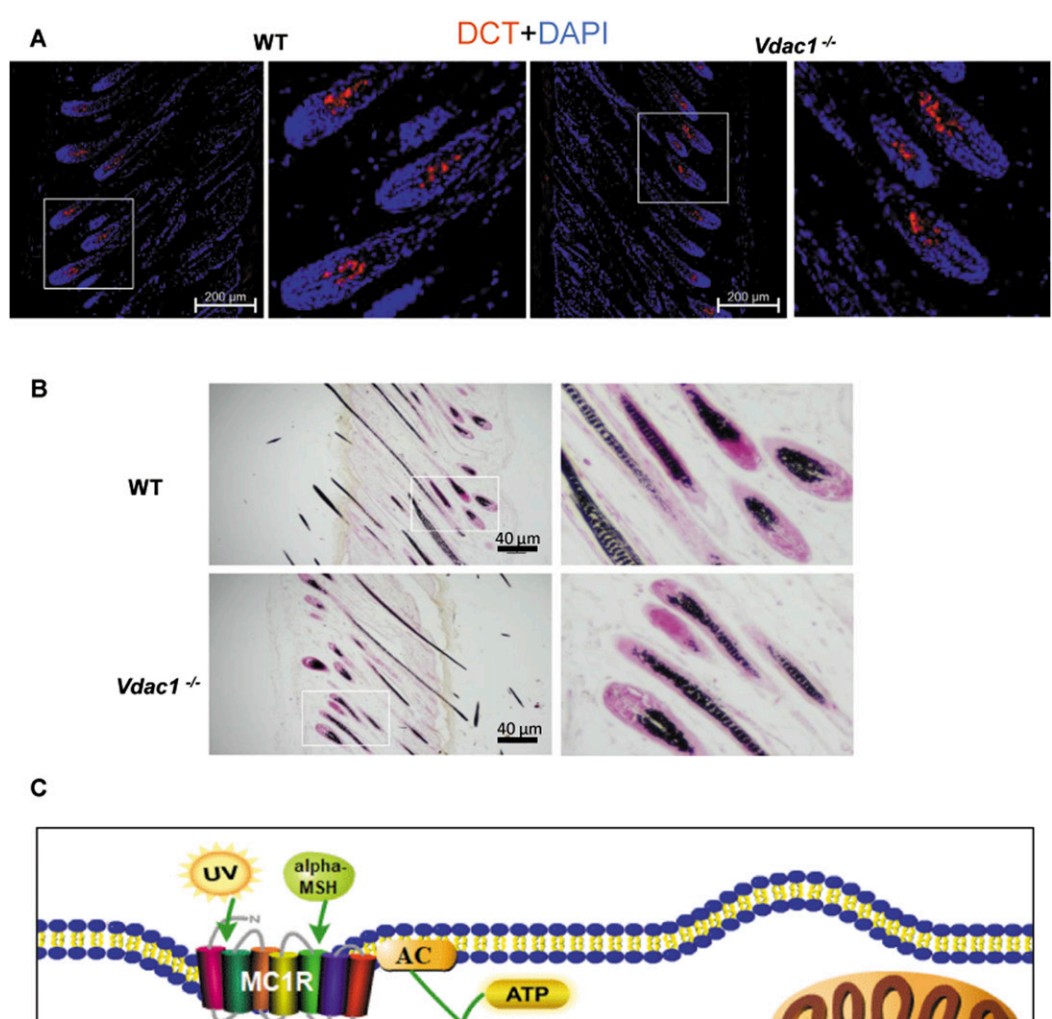

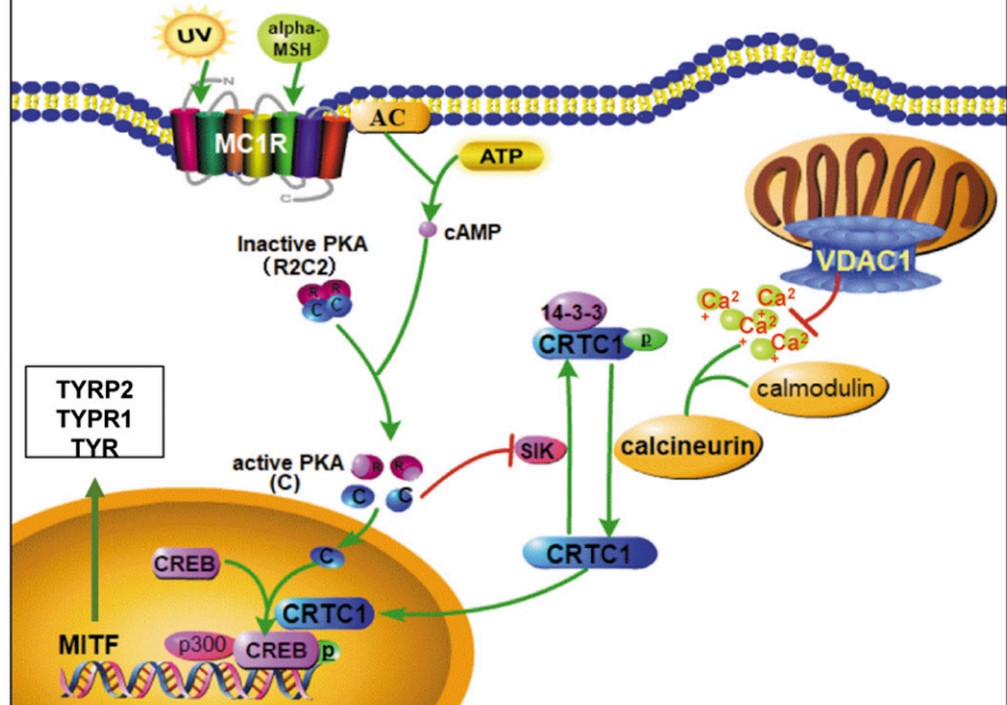

**Figure 7. Melanocyte number is unchanged in *Vdac1*-knockout mice and a proposed model.**
**(A)** The melanocyte number was not affected in *Vdac1*-knockout mice. The skin sections of wild-type (WT) and *Vdac1*$^{-/-}$ mice were stained with antibody to DCT. Enlarged images of the boxed area are shown in the right. Scale bar: 200 $\mu$m. **(B)** Representative images of Fontana-Masson staining of melanin in mouse skin. Enlarged images of the boxed area are shown in the right panel. No apparent change of melanocytes is observed in *Vdac1*$^{-/-}$ mice compared with the WT mice. Scale bar: 40 $\mu$m. **(C)** A proposed model depicting the negative regulation of VDAC1 in melanogenesis. In melanocytes, the depletion of mitochondrial VDAC1 increases the concentration of free Ca$^{2+}$ in the cytoplasm, thereafter activates calcineurin. Activated calcineurin dephosphorylates p-CRTC1 and therefore CRTC1 is released from 14-3-3 protein in the cytoplasm. CRTC1 is then translocated to the nucleus, where MITF transcription is activated through the coordination with p-CREB. As a result, melanogenesis is up-regulated.

pre-warmed ammoniacal silver solution for 30 min at 56°C. The slices were washed with distilled water for three times. Then the slices were treated with hypo solution for 3 min and washed with distilled water for four times. The slices were stained with Neutral Red Staining Solution for 5 min and washed for 1 min. Then the slices were dehydrated and mounted in synthetic resin. Pictures were captured with a Nikon Ci-L microscope (Nikon) at 100× magnification.

### Mice husbandry

The *Vdac1*-knockout mice (*Vdac1$^{-/-}$*) were generated through CRISPR-Cas9 in the C57BL/6J background by Beijing BIOCYTOGEN's EGE system. All mice studies were approved by the Institutional Animal Care and Use Committee of Institute of Genetics and Developmental Biology, Chinese Academy of Sciences (mouse protocol # AP2021028). To ensure the genotypes of the littermates, PCR amplifications were designed according to the targeted deletion of exons 2 and 8. Primers used for genotyping are listed in Table S4. *Vdac1$^{+/+}$* from the littermates was used as the background control. Only male mice were used for experiments to prevent variability in melanin caused by estrogen.

### Data analysis

The Student unpaired two-tailed *t* test was used for statistical analyses. A *P*-value less than 0.05 was considered as statistically significant.

## Data Availability

All data (including RT–PCR) that support the findings of this study are available upon reasonable request from the corresponding author.

## Supplementary Information

## Acknowledgments

This work was supported by grants from the Ministry of Science and Technology of China (2019YFA0802104 [to W Li]), from the National Natural Science Foundation of China (31900549 [J Gong]; 31830054 [to W Li]; 91954000 [to. W Li]). We thank Dr. Ting Chen for kindly providing the antibodies to DCT used in our immunofluorescence staining. We thank Dr. Kaiju Jiang for providing technical suggestions to immunofluorescence staining of mouse skin. We are grateful to Xueke Tan, Zhongshuang Lv, and Xixia Li for helping with electron microscopy sample preparation at the Center for Biological Imaging (CBI), Institute of Biophysics, Chinese Academy of Science.

### Author Contributions

J Wang: investigation and writing—original draft.

J Gong: data curation, investigation, and writing—original draft.

Q Wang: methodology.

T Tang: supervision.

W Li: conceptualization, data curation, supervision, project administration, and writing—review and editing.

### Conflict of Interest Statement

The authors declare that they have no conflict of interest.

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
