## [Reviewer comments · Life Science Alliance]

Life Science Alliance

VDAC1 negatively regulates melanogenesis through the Ca²⁺-calcineurin-CRTC1-MITF pathway

Jianli Wang, Juanjuan Gong, Qiao-Chu Wang, Tie-Shan Tang, and Wei Li

DOI: <https://doi.org/10.26508/lsa.202101350>

Corresponding author(s): Wei Li, Capital Medical University

Review Timeline:

Submission Date:	2021-12-23
Editorial Decision:	2022-02-07
Revision Received:	2022-04-16
Editorial Decision:	2022-05-18
Revision Received:	2022-05-21
Accepted:	2022-05-23

Transaction Report:

February 7, 2022

Re: Life Science Alliance manuscript #LSA-2021-01350-T

Prof. Wei Li
Capital Medical University
Beijing Children's Hospital
56 Nan-Li-Shi Road
Xicheng District, Beijing 100045
China

Dear Dr. Li,

Thank you for submitting your manuscript entitled "VDAC1 negatively regulates melanogenesis through the Ca²⁺-calcineurin-CRTC1-MITF pathway" to Life Science Alliance. The manuscript was assessed by expert reviewers, whose comments are appended to this letter. We, thus, encourage you to submit a revised version of the manuscript back to LSA that responds to all of the reviewers' points.

Thank you for this interesting contribution to Life Science Alliance. We are looking forward to receiving your revised manuscript.

Sincerely,

B. MANUSCRIPT ORGANIZATION AND FORMATTING:

Reviewer #1 (Comments to the Authors (Required)):

Overall, the manuscript presents important work suggesting a role for calcium homeostasis in melanocyte function. However, there are a number of structural issues that need to be revised before considering for publication.

1. Needs a grammar check for numerous small English language syntax issues (verb conjugations, punctuation, sentence structure).
2. P4para2: The prior study showed indirect evidence of a relationship between mitochondrial calcium efflux and mitochondrial efflux. The assertion that the previous work "showed that NCKX5, located in the mitochondria, transports mitochondrial [calcium] to the melanosome" is stronger than the evidence, which remains a hypothesis at this point. There is a good review in Le, et al, PMID: 34021746
3. Future work might include assessing mutations in SLC24A5 said to cause albinism in people. Published photographs of affected individuals show dramatic hair and skin pigment changes not seen in mouse models.
4. Many of the significance values are modest, for instance $p=0.041$ in Figure 1F. Where there appears to be only one dot on the normal control (NC?), should it be presumed that the error bars were too small to be seen? If so, a note to that effect should be added. Figure numbers need to be added to the figures (at least in the combined review document). Some guessing was required to figure out how to map the non-consecutive numbering in the figure legends (see figure two legend) with the figure elements. It was difficult to navigate without that data. This concern may not be an issue in the final printed version, however.
5. Figure 4A does not make sense with very significant P values and overlapping 1 s.d. bars. The authors should make sure that those calculations were correct
6. The mouse photographs are poorly lit and would benefit from the inclusion of a color scale. The authors note that there is no difference in coal color, but there does appear to be one from the photographs.

Reviewer #2 (Comments to the Authors (Required)):

The manuscript by Wang and collaborators offers an innovative idea about a possible link between mitochondrial functionality and melanogenesis.

The data are well presented, however there are several important aspects that might be improved.

Major points:

-The choice of the model used is not clear. The correct model for human melanogenesis is human melanocyte cell culture rather than melanoma lines that exhibit many pathological-conditioned intracellular signalings (MAPKs, Wnt signaling.....). Also, the metabolic profile of melanoma cells is frequently different from normal cells, a situation that potentially impacts mitochondria biology.

-Furthermore, the validity of the data obtained with the melanoma lines is not confirmed by the in vivo model. This negative figure is not adequately considered in the last paragraph of the results section and in general discussion of the data.

Therefore, I believe that in the absence of experiments performed on low-pass human melanocytes or ex vivo human skin biopsies, the work substantially loses interest.

Minor points:

The introduction is too short and non-exhaustive and accompanied by a few often quite old references.

Es. "Thus, investigating the regulation of melanogenesis not only elucidates the mechanisms of pigmentary disorders, but also provides new intervention clues to these diseases (Yamaguchi & Hearing, 2009)."

-MSH is referred to alpha-MSH? Please use the correct term.

-The discussion lacks a consideration on the pathological implication in conditions of hyper or hypo-pigmentation that could enrich this manuscript.

Reviewer #3 (Comments to the Authors (Required)):

Wang J, et al's manuscript entitled "VDAC1 negatively regulates melanogenesis through the Ca²⁺-calcineurin-CRTC1-MITF pathway" demonstrated the role of VDAC1 in melanogenesis in vitro and in vivo. This story is interesting. A new mouse model was generated. Although the melanosome concentration was found higher in VDAC1^{-/-} mice, it remains unclear melanin production was from more melanin production, higher melanocyte concentration, or both. It's necessary to identify the melanocyte concentration and melanin production in ear epidermis or tail epidermis. It's also useful to identify the size of melanosome.

Point-to-Point Replies

Reviewer #1 (Comments to the Authors (Required)):

Overall, the manuscript presents important work suggesting a role for calcium homeostasis in melanocyte function. However, there are a number of structural issues that need to be revised before considering for publication.

1. Needs a grammar check for numerous small English language syntax issues (verb conjugations, punctuation, sentence structure).

Response: We have checked the manuscript carefully to correct the typos and errors in grammar and spelling. The changes are marked in red in the revised manuscript.

2. P4para2: The prior study showed indirect evidence of a relationship between mitochondrial calcium efflux and mitochondrial flux. The assertion that the previous work "showed that NCKX5, located in the mitochondria, transports mitochondrial [calcium] to the melanosome" is stronger than the evidence, which remains a hypothesis at this point. There is a good review in Le, et al, PMID: 34021746

Response: To weaken the statement, we revised the sentence as "*Our previous study showed that NCKX5, located in the mitochondria, may play an important role in regulating melanosomal Ca²⁺ homeostasis that is required for pigment production (Le et al, 2021; Zhang et al, 2019)*". PMID: 34021746 has been cited in this revision.

Le L, Sires-Campos J, Raposo G, Delevoye C, Marks MS (2021) Melanosome Biogenesis in the Pigmentation of Mammalian Skin. *Integr Comp Biol* **61**: 1517-1545

Zhang Z, Gong J, Sviderskaya EV, Wei A, Li W (2019) Mitochondrial NCKX5 regulates melanosomal biogenesis and pigment production. *J Cell Sci* **132**: jcs232009

3. Future work might include assessing mutations in SLC24A5 said to cause albinism in people. Published photographs of affected individuals show dramatic hair and skin pigment changes not seen in mouse models.

Response: We agree with the reviewer's comment. We are studying the details of how SLC24A5 affects melanogenesis.

4. Many of the significance values are modest, for instance p=0.041 in Figure 1F. Where there appears to be only one dot on the normal control (NC?), should it be presumed that the error bars were too small to be seen? If so, a note to that effect should be added. Figure numbers need to be added to the figures (at least in the combined review document). Some guessing was required to figure out how to map the non-consecutive numbering in the figure legends (see figure two legend) with the figure elements. It was difficult to navigate without that data. This concern may not be

an issue in the final printed version, however.

Response: There are two ways to calculate the relative expression level: 1) all the values are normalized to the mean value of the normal control (NC); 2) the values of each experiment are normalized to the value of the NC, and the value of all NC groups from independent experiments will be 1 (Adams et al, 2020; Liu et al, 2020; Zhang et al, 2018). The significance values will not be affected by these two methods. When we processed our data using the second method, we normalized the data by value of NC which was 1. Therefore, there is only one dot to represent three independent experiments in the NC group in Figure 1F. The figure numbers have been added in the revised figures.

Adams CL, Ercolano E, Ferluga S, Sofela A, Dave F, Negroni C, Kurian KM, Hilton DA, Hanemann CO (2020) A Rapid Robust Method for Subgrouping Non-NF2 Meningiomas According to Genotype and Detection of Lower Levels of M2 Macrophages in AKT1 E17K Mutated Tumours. *Int J Mol Sci* **21**: 1273

Liu Y, Zou GJ, Tu BX, Hu ZL, Luo C, Cui YH, Xu Y, Li F, Dai RP, Bi FF, Li CQ (2020) Corticosterone Induced the Increase of proBDNF in Primary Hippocampal Neurons Via Endoplasmic Reticulum Stress. *Neurotox Res* **38**: 370-384

Zhang X, Yu D, Geng H, Li F, Lv L, Zhao L, Yan C, Li B (2018) Dual effects of arsenic trioxide on tumor cells and the potential underlying mechanisms. *Oncol Lett* **16**: 3812-3820

5. Figure 4A does not make sense with very significant P values and overlapping 1 s.d. bars. The authors should make sure that those calculations were correct.

Response: The basal calcium level varies from different cells. When detecting the cytosolic calcium level, the numbers of cells from each group were about 100. The large sample size of each group ensured the very significant P values. The final data was from two independent experiments. Therefore, the total sample size of each group was 252, 280, and 253, respectively. These numbers have been added in the Figure 4A legends. The P values were calculated by Graphpad prism. We confirmed that the statistics was correct.

6. The mouse photographs are poorly lit and would benefit from the inclusion of a color scale. The authors note that there is no difference in coal color, but there does appear to be one from the photographs.

Response: We thank for the reviewer's comment. We have re-taken the mouse photographs by comparing the color of the dorsal fur, ventral fur and toes, ears and tail of three different genotypes (+/+, +/-, -/-). There was no apparent difference in the color of these mice. The revised picture is shown in Figure 5E.

Reviewer #2 (Comments to the Authors (Required)):

The manuscript by Wang and collaborators offers an innovative idea about a possible link between mitochondrial functionality and melanogenesis. The data are well presented, however there are several important aspects that might be improved.

Major points:

-The choice of the model used is not clear. The correct model for human melanogenesis is human melanocyte cell culture rather than melanoma lines that exhibit many pathological-conditioned intracellular signalings (MAPKs, Wnt signaling.....). Also, the metabolic profile of melanoma cells is frequently different from normal cells, a situation that potentially impacts mitochondria biology.

Response: We really appreciate the reviewer's thoughtful comments. To avoid the interference of pathological-conditioned intracellular signaling in melanoma MNT1 cell, we repeated the experiment of melanin content (Figure 1E, F), the protein level of TYR/TYRP1/TYRP2 (Figure 2E-H) in melan-a, a kindly gift from Dorothy C. Bennett (University of London). Melan-a is a line of pigmented melanocytes, derived from normal epidermal melanoblasts from embryos of inbred C57BL mice. Melan-a cells have the diploid chromosome number and do not form tumors in syngeneic or nude mice (Bennett et al, 1987). We used melan-a for melanocyte cell culture model in our work to exclude the possible influence by pathological-condition in MNT1 cells. In addition, our results shown in Figure 6 also reproduced the higher expression of melanogenic genes in the *Vdac1^{-/-}* mice, which supports that our proposed VDAC1-regulated melanogenesis pathway is functional.

In addition, we agree that human melanocyte cell culture will be more suitable for our work as the reviewer suggested, so we have ordered it from a USA company. However, due to the COVID-19 pandemic, the delivery time is uncertain and could be too long to maintain the cell properly. We hope to get the human melanocyte cell line as soon as the situation improves. Nevertheless, our current data suggest that the melanogenesis pathway in this study is not due to the side effects of tumorigenic pathways.

Bennett DC, Cooper PJ, Hart IR (1987) A line of non-tumorigenic mouse melanocytes, syngeneic with the B16 melanoma and requiring a tumour promoter for growth. *Int J Cancer* **39**: 414-418

-Furthermore, the validity of the data obtained with the melanoma lines is not confirmed by the in vivo model. This negative figure is not adequately considered in the last paragraph of the results section and in general discussion of the data. Therefore, I believe that in the absence of experiments performed on low-pass human melanocytes or ex vivo human skin biopsies, the work substantially loses interest.

Response: We have added the following sentences in the last paragraph of Result:

"However, there was no obvious change in melanin content (Fig. 5E, F) and melanocyte number between *Vdac1*^{-/-} mice and controls (Fig.7A, B). The complex signals that affect melanogenesis in vivo, such as the endogenous hormone α -MSH, may cover up the effect caused by *Vdac1* depletion. Thus, our work provides a new melanogenic mechanism in a UVB- or α -MSH-independent manner."

Our *in vivo* model did show the increase of several melanogenic genes (Figure 6) although the coat color changes were not obvious in the C57BL/6J mice. One explanation to this puzzle could be due to the α -MSH-cAMP-PKA pathway that contributes to the production of melanin, which may substitute for the effects of the Ca²⁺-CaN-CRTC1-MITF pathway on melanin production *in vivo*. We also have added the following sentences in Discussion: "Likewise, the increased number of melanosomes in RPE may also be compensated by the decreased size of melanosomes, without much increase in melanin content although the melanogenic genes are upregulated. In addition, the *Vdac1*^{-/-} were generated in the C57BL/6J background. The C57BL/6J mice is black, which may make it difficult to distinguish whether the color of mice becomes much darker when *Vdac1* is deficient. The inbred strain mice with agouti appearance may be another choice. The *Vdac1*^{-/-} mice in the C57BL/6J background will be transferred to the agouti background to observe possible color changes in vivo in our future work."

In summary, we observed that the melanin content was increased after VDAC1 knockdown in MNT and melan-a cells in a Ca²⁺-calcineurin-CRTC1-MITF dependent manner. The number of melanosomes was increased, but the size of melanosomes was decreased, suggesting that melanosome biogenesis is compromised in *Vdac1* knockout mice. Consistently, the mRNA levels of *Mitf*, *Tyr*, *Tyrp1*, and *Tyrp2* were increased in *Vdac1* knockout mice. However, there was no obvious difference in melanin content in the mouse skin after *Vdac1* was knocked out. We suspected that endogenous pathways such as the α -MSH-cAMP-PKA pathway may substitute for the production of melanin, which may cover up the effects of the disrupted Ca²⁺-CaN-CRTC1-MITF pathway on melanin production *in vivo*. The decreased size of melanosomes may also have a compensatory role for the increased number of melanosomes as shown in the RPE. In addition, the *Vdac1* KO mice were generated in the C57BL/6J background. The C57BL/6J mice is black, which may make it difficult to distinguish whether the color of mice become much darker after VDAC1 knockout.

Minor points:

The introduction is too short and non-exhaustive and accompanied by a few often quite old references. Es. "Thus, investigating the regulation of melanogenesis not only elucidates the mechanisms of pigmentary disorders, but also provides new intervention clues to these diseases (Yamaguchi & Hearing, 2009)."

Response: We have changed some old references with several recent references (Le et al, 2021; Li et al, 2022; Yun et al, 2019).

We have added the following statements in the revised Introduction.

"It has been shown that blocking the nuclear import of CRTC1 inhibits melanin

production, suggesting that CRTC1 is a potential target in the treatment of pigmentary disorders (Yun et al, 2019)."

"VDAC1 is a multi-functional channel mediating the entry of metabolites (e.g., NADH, pyruvate, malate, succinate, and nucleotides) into the mitochondria, and the exit of newly formed molecules such as ROS into the cytosol (Shoshan-Barmatz et al, 2018b)."

"Melanin is synthesized within the melanosomes, a type of lysosome-related organelles (LROs) (Li et al, 2021). Melanosome biogenesis/maturation is also regulated by Ca²⁺ homeostasis in the melanosome."

Le L, Sires-Campos J, Raposo G, Delevoye C, Marks MS (2021) Melanosome Biogenesis in the Pigmentation of Mammalian Skin. *Integr Comp Biol* **61**: 1517-1545

Shoshan-Barmatz V, Nahon-Crystal E, Shteinifer-Kuzmine A, Gupta R (2018b) VDAC1, mitochondrial dysfunction, and Alzheimer's disease. *Pharmacological research* **131**: 87-101

Li W, Hao CJ, Hao ZH, Ma J, Wang QC, Yuan YF, Gong JJ, Chen YY, Yu JY, Wei AH (2022) New insights into the pathogenesis of Hermansky-Pudlak syndrome. *Pigment Cell Melanoma Res*. doi: 10.1111/pcmr.13030.

Tian X, Cui Z, Liu S, Zhou J, Cui R (2021) Melanosome transport and regulation in development and disease. *Pharmacol Ther* **219**: 107707

Yun CY, Hong SD, Lee YH, Lee J, Jung DE, Kim GH, Kim SH, Jung JK, Kim KH, Lee H, Hong JT, Han SB, Kim Y (2019) Nuclear Entry of CRTC1 as Druggable Target of Acquired Pigmentary Disorder. *Theranostics* **9**: 646-660

-MSH is referred to alpha-MSH? Please use the correct term.

Response: Yes. This has been corrected in the revised manuscript.

-The discussion lacks a consideration on the pathological implication in conditions of hyper or hypo-pigmentation that could enrich this manuscript.

Response: We appreciate the reviewer's suggestion and we have added the following sentences in the revised Discussion: "In addition, the aberrant melanin production due to abnormal regulation of VDAC1 may be implicated in pathological hyper or hypo-pigmentation, such as albinism, freckles, or even life-threatening melanoma. Our work provides a potential target for these disorders."

Reviewer #3 (Comments to the Authors (Required)):

Wang J, et al's manuscript entitled "VDAC1 negatively regulates melanogenesis through the Ca²⁺-calcineurin-CRTC1-MITF pathway" demonstrated the role of VDAC1 in melanogenesis *in vitro* and *in vivo*. This story is interesting. A new mouse model was generated. Although the melanosome concentration was found higher in VDAC1^{-/-} mice, it remains unclear melanin production was from more melanin production, higher melanocyte concentration, or both. It's necessary to identify the melanocyte concentration and melanin production in ear epidermis or tail epidermis. It's also useful to identify the size of melanosome.

Response: We appreciate the reviewer's suggestion. We have performed additional experiments to address these points. In our first manuscript, we showed that there was no obvious difference in melanin content between WT and *Vdac1*^{-/-} mice, but the melanosome number was higher in *Vdac1*^{-/-} mice than that in WT mice. The melanocyte number is also another important aspect that should be detected. As the reviewer suggested, we performed immunostaining with anti-DCT antibody on formalin-fixed paraffin-embedded skin sections. Melanocytes in skin hair follicles were no significantly different between the two groups (Figure 7A). To further detect the melanin concentration, Fontana-Masson staining was performed. The data revealed that there was no significant difference in melanin pigment in the mouse skin between the two groups (Figure 7B).

In our first manuscript, we showed that the number of melanosomes was increased in the *Vdac1*^{-/-} mice. Following the reviewer's suggestion, we also quantified the sizes of the melanosomes. The data showed that the sizes of melanosomes in WT mouse were larger than that in KO mice. This may compensate the increase number of melanosomes without much increase of melanin production *in vivo*. We have included this data in Figure 6C.

May 18, 2022

RE: Life Science Alliance Manuscript #LSA-2021-01350-TR

Prof. Wei Li
Capital Medical University
Beijing Children's Hospital
56 Nan-Li-Shi Road
Xicheng District, Beijing 100045
China

Dear Dr. Li,

Thank you for submitting your revised manuscript entitled "VDAC1 negatively regulates melanogenesis through the Ca²⁺-calcineurin-CRTC1-MITF pathway". We would be happy to publish your paper in Life Science Alliance pending final revisions necessary to meet our formatting guidelines.

- please add ORCID ID for corresponding author-you should have received instructions on how to do so
- please add the Twitter handle of your host institute/organization as well as your own or/and one of the authors in our system
- please use the [10 author names, et al.] format in your references (i.e. limit the author names to the first 10)
- we encourage you to introduce your panels in alphabetical order in the figure legends
- please add a Data Availability statement with accession information for the RT-PCR data

A. FINAL FILES:

B. MANUSCRIPT ORGANIZATION AND FORMATTING:

**Submission of a paper that does not conform to Life Science Alliance guidelines will delay the acceptance of your

manuscript.**

The license to publish form must be signed before your manuscript can be sent to production. A link to the electronic license to publish form will be sent to the corresponding author only. Please take a moment to check your funder requirements.

Sincerely,

Reviewer #3 (Comments to the Authors (Required)):

All my concerns have been addressed.

May 23, 2022

RE: Life Science Alliance Manuscript #LSA-2021-01350-TRR

Prof. Wei Li
Capital Medical University
Beijing Children's Hospital
56 Nan-Li-Shi Road
Xicheng District, Beijing 100045
China

Dear Dr. Li,

Thank you for submitting your Research Article entitled "VDAC1 negatively regulates melanogenesis through the Ca²⁺-calcineurin-CRTC1-MITF pathway". It is a pleasure to let you know that your manuscript is now accepted for publication in Life Science Alliance. Congratulations on this interesting work.

DISTRIBUTION OF MATERIALS:

Again, congratulations on a very nice paper. I hope you found the review process to be constructive and are pleased with how the manuscript was handled editorially. We look forward to future exciting submissions from your lab.

Sincerely,
